# Solvent Effects in the Preparation of Catalysts Using Activated Carbon as a Carrier

**DOI:** 10.3390/nano13030393

**Published:** 2023-01-18

**Authors:** Zhuang Xu, Mengli Li, Guowang Shen, Yuhao Chen, Dashun Lu, Peng Ren, Hao Jiang, Xugen Wang, Bin Dai

**Affiliations:** 1School of Chemistry and Chemical Engineering, Shihezi University, Shihezi 832000, China; 2Key Laboratory for Green Processing of Chemical Engineering of Xinjiang Bingtuan, Shihezi 832000, China

**Keywords:** solvent effect, activated carbon, catalysts preparation

## Abstract

The role of solvents is crucial in catalyst preparation. With regard to catalysts prepared with activated carbon (AC) as the carrier, when water is used as a solvent it is difficult for the solution to infiltrate the AC. Because AC comprises a large number of C atoms and is a nonpolar material, it is more effective for the adsorption of nonpolar substances. Since the water and active ingredients are polar, they cannot easily infiltrate AC. In this study, the dispersion of the active component was significantly improved by optimizing the solvent, and the particle size of the active component was reduced from 33.08 nm to 15.30 nm. The specific surface area of the catalyst is significantly increased, by 10%, reaching 991.49 m^2^/g. Under the same reaction conditions, the conversion of acetic acid by the catalyst prepared with the mixed solvent was maintained at approximately 65%, which was 22% higher than that obtained using the catalyst prepared with water as the solvent.

## 1. Introduction

Activated carbon (AC) comprises a large number of C atoms and is widely used as a nonpolar material [1]. It is generally used to remove [2], recover [3] and separate various compounds [4] from the gas and liquid phases. AC is also commonly used as a carrier in industrial production catalysts [5,6], and the preparation process generally involves impregnation to load the active component to AC. In the preparation of catalysts using AC as a carrier, the active ingredient is first dissolved in water to form a solution of a certain concentration and then added to the carrier. In the impregnation process, water and the active component are polar [7], and AC is nonpolar [8]). According to the principle of increased solubility with similar structure [9], AC has a higher adsorption capacity for nonpolar materials and a lower adsorption capacity for polar molecules, such as salts or water. Because of the complex adsorption process, the dispersion of active components on AC and the size of active sites are closely related to the performance of the catalyst [10]. During impregnation adsorption, the polarity of AC substantially differs from the polarity of the active component salt solution [11], so the active component on AC can suffer from uneven dispersion [12], particle agglomeration [13], and the blockage of catalyst pore channels [14], affecting catalyst performance.

The solvent plays a key role in catalyst preparation as a bridge between the active component and the carrier. Studies by Wei-Jing Li et al. [15] show that using isopropanol as the solvent governs the catalyst particle size with a superior dispersion and a high catalytic activity. Min-Su Jang et al. [16] developed a new and facile preparation method for egg-shell catalysts using immiscibility between hydrophobic solvent preoccupied inside pellet and aqueous hydrophilic precursor solution. Mu et al. [17] synthesized RuRh_2_ bimetallene nanoring with abundant structural defects using a mixed solvent strategy, showing high-quality active hydrogen evolution reaction performance, which is significantly higher than that of the commercial Pt catalyst. Cavuoto [18] affected the activity and selectivity of the catalyst by adjusting the solvent. The Hutchings [19] group improved the dispersion of the active ingredient and the activity of the catalyst using low-polarity and low-boiling-point solvents. Bacariza et al. [20] changed the dispersion of active components by adjusting the solvent.

The most industrially mature and widely used catalyst [21] in the acetylene process for the synthesis of vinyl acetate (Vac) is the zinc acetate-loaded AC (Zn(OAc)_2_/AC) catalyst. Water is used as the solvent in the preparation of this catalyst. Since the active component and water are polar materials but AC is a nonpolar material, the active component is nonuniformly dispersed in the catalyst. To solve this problem, we used a mixed solvent to reduce the polarity of the solution and improve the dispersion of the active component, which, in turn, improved the activity of the catalyst.

## 2. Experimental Section

### 2.1. Materials

The materials used in this study were carbon powder, methanol (Tianjin Fuyu, Tianjin, China, AR), ethanol (Tianjin Fuyu, Tianjin, China, AR), n-Propanol (Tianjin Fuyu, Tianjin, China, AR), isopropanol (Tianjin Fuyu, Tianjin, China, AR), Zn (NO_3_)_2_·6H_2_O (Tianjin Guangfu, Tianjin, China, 98%), urea (Tianjin Guangfu, Tianjin, China, AR), AC (pH = 6–8, coconut carbon), acetylene (gas, 99.99%), acetic acid (Tianjin Beilian, Tianjin, China, 99.8%), and deionized water.

### 2.2. Preparation of Catalysts

Mixed solvents A, B, C, and D were prepared by mixing methanol, ethanol, n-propanol, and isopropanol (20 wt%) with deionized water (80 wt%), respectively.

A certain amount of 40–60 mesh coconut shell carbon (AC) was weighed using an analytical balance, and impurities and ash were removed from AC. An amount of 1.00 g AC, 0.50 g urea, and 0.50 g Zn (NO_3_)_2_ were weighed and placed into each of five 50-mL beakers. A total of 30 mL deionized water, 30 mL mixed solvent A, 30 mL mixed solvent B, 30 mL mixed solvent C, and 30 mL mixed solvent D were added to these beakers. After placing the magnets, the solution was placed on a magnetic stirring device, the beakers were sealed with cling film, the stirring rate was set to 600 rpm/min, and the solution was stirred for 12 h at room temperature. Consequently, the stirred solution was put into a blast drying oven and dried overnight at 80 °C. The dried drug was taken out from the oven, and the drug was treated with the facile carbon bath method [22] by setting the temperature to 800 °C, holding it at 800 °C for 2 h, and then cooling it down. Then, the muffle furnace naturally cooled to room temperature. The quantity of the high-temperature-treated drug was weighed, and the catalyst performance was tested after acetic acid activation.

### 2.3. Characterization

X-ray photoelectron spectroscopy (XPS) data were recorded by Thermo ESCALAB 250Xi electron spectrometer (Thermo Fisher Scientific, Waltham, MA, USA). Transmission electron microscopy (TEM) and high-resolution transmission electron microscopy (HRTEM) images were collected by Tecnai G2 F30 S-TWIN (FEI, Portland, OR, USA) instrument. Nitrogen adsorption-desorption isotherms were carried out on a Micromeritics ASAP 2460 apparatus (Micrometrics Instrument Corporation, Atlanta, GA, USA). The specific surface areas, total pore volume, and average pore size were calculated based on the Brunauer–Emmett–Teller (BET) and Barrett–Joyner–Halenda (BJH) methods [23]. The inductively coupled plasma (ICP) evaluation was performed using the Agilent ICPOES 730 (Agilent Technologies, Palo Alto, CA, USA) for metal elemental measurement. The temperature-programmed desorption (TPD, Micromeritics ChemiSorb 2720, Norcross, GA, USA) was performed by heating the sample from 25 °C to 800 °C with a ramp rate of 10 °C/min in N_2_ atmosphere.

### 2.4. Heterogeneous Catalytic Evaluation

The catalyst was tested in a homemade fixed-bed reactor with an inner diameter of 10 mm and a length of 50 cm. An amount of 2 mL of the catalyst was filled into the reactor tube and fixed on the heating furnace. The top of the reaction tube was connected to the feed gas line, and the bottom of the reaction tube was connected to the condenser inlet. The N_2_ control valve was opened, the valve opening was adjusted, and the gas tightness of all pipelines was tested. The heating furnace (T = 200 °C) and the acetic acid vaporization device (T = 150 °C) were turned on to heat the device. The peristaltic pump was turned on 30 min after N_2_ purging, and CH_3_COOH was blown into the reaction tube by nitrogen through the vaporization device. The catalyst was activated with CH_3_COOH, while the furnace temperature was adjusted to 220 °C (the purpose of adjusting the temperature in parts was to prevent damage to the catalyst caused by the rapid temperature rise). After 30–40 min of activation with CH_3_COOH, the N_2_ valve was closed, and the C_2_H_2_ valve was opened. GHSV_C2H2_ = 500 h^−1^, and the feed molar ratio C_2_H_2_ (g)/CH_3_COOH (g) = 3. Subsequent samples were collected every 1 h, and the products were tested and analyzed using gas chromatography.

## 3. Results and Discussion

### 3.1. Performance Evaluation of Catalysts

Five different catalysts were prepared using water and mixed solvents A, B, C, and D, and all the other conditions were the same. The catalyst performance is shown in Figure 1a. The conversion of acetic acid was maintained at approximately 43% when pure water was used as the solvent. The catalyst activity was enhanced by replacing pure water with the mixed solvents. The catalysts prepared using mixed solvent B showed a stable acetic acid conversion of 60%, probably because methanol reduced the polarity of the solution during the preparation process [24], leading to an increase in catalyst dispersion. The catalysts prepared with mixed solvents C and D, using n-propanol and isopropanol, respectively, showed the similar catalytic performance, which might be attributed to the proximity of their molecular structures. The catalysts prepared with mixed solvent A maintained an acetic acid conversion of approximately 65%, a 22% increase compared with the catalyst prepared with water as the solvent.

We ranked the polarity of alcohols that improved the polarity of the solvents as water > mixed solvent A > mixed solvent B > mixed solvent C > mixed solvent D (Figure 1b). The fitted linear relationship in the figure indicates that the acetate conversion and the polarity of the solvent molecules are closely related. As the polarity of the solvent increases, the activity of the catalyst first increases and then decreases. The polarity of the solvent affects the dispersion of the active substance on the carrier and its interaction with the carrier. This may be because AC is a hydrophobic material, making it difficult for Zn(OAc)_2_ solution to enter. However, a weakly polar solvent has better wettability for AC [25], which, in turn, allows for better dispersion of the active substance on the carrier.

### 3.2. Analysis of Contact Angle

Substances with similar structures dissolve each other. Water is a polar material, whereas AC is a nonpolar material [26]; thus, AC is not dissolved well in solutions prepared using water. To demonstrate that the reduction in solution polarity can make AC better wetted by the solution, contact angle tests were conducted for water and four mixed solvents using AC as the substrate, as shown in Figure 2. The contact angle between pure water and AC was 129.64°, indicating that AC is a hydrophobic material and is nonpolar. The contact angle of mixed solvent A with AC was 80.9°, the contact angle of mixed solvent B with AC was 67.53°, the contact angle of mixed solvent C with AC was 23.28°, and the contact angle of mixed solvent D with AC was 17.20°. The polarity of solvents obtained using contact angle tests decreased in the order of water > mixed solvent A > mixed solvent B > mixed solvent C > mixed solvent D.

### 3.3. Relationship between Solvent Polarity and Active Components

The contact angle results showed that the contact angle between the solution and AC decreased as the polarity of the solvent decreased, indicating that the solution became easier to infiltrate into the interior of AC. However, mixed solvent A, instead of mixed solvent D, showed the best reaction performance because the loading of the active component affects the performance of the catalyst. Thus, we tested the loading of the active component, as shown in Figure 3. The loading of Zn(OAc)_2_ first increased and then decreased as the polarity of the solvent decreased. The decrease in polarity did increase the loading of Zn(OAc)_2_, but when the polarity substantially decreased, the loading of Zn(OAc)_2_ decreased instead. In terms of material properties, since both water and Zn(OAc)_2_ are polar materials and Zn(OAc)_2_ has the highest solubility in water, the solubility of Zn(OAc)_2_ was reduced with the decreased polarity of the solution, so mixed solvent A showed the best reactivity.

### 3.4. TEM Mapping Analysis

After replacing pure water with the mixed solvent, the active component infiltration of AC became easier. To further illustrate the enhancement of catalyst activity owing to mixed solvent A, TEM mapping was conducted on the catalysts prepared with water and mixed solvent A. Figure 4a shows the TEM mapping of the catalyst prepared with water, and Figure 4b shows the TEM mapping of the catalyst prepared with mixed solvent A. Figure 4a shows that Zn agglomerated on the surface of AC, forming poorly dispersed large particles. Because the solution polarity is high when the catalyst is prepared with water and AC is a nonpolar material, the solution infiltration of AC is less effective. On the other hand, in Figure 4b, the size of Zn particles on the surface of AC was significantly reduced, and Zn was more uniformly dispersed. Due to the reduction in solution polarity, the solution infiltration of AC is easier, and the active component can be dispersed more easily.

To verify that the introduction of the solvent mixture improved the dispersion of active components, the particle sizes of the active components in the TEM images were calculated (Figure 5). Figure 5a shows the catalyst prepared with water, and the average size of Zn particles calculated by the software was 33.08 ± 13.72 nm. Zn particles were nonuniformly dispersed, and particle agglomeration was observed. The average size of Zn particles was 15.30 ± 4.99 nm for the catalyst prepared using mixed solvent A (Figure 5b). The change in the polarity of the solvent significantly reduced the particle size of Zn and improved the dispersion of the active component, consistent with the TEM mapping results. The catalysts prepared using mixed solvent A showed a significant decrease in the particle size of Zn and a significant increase in the dispersion and loading of Zn, which may be responsible for the elevated catalyst performance [27].

### 3.5. Analysis of BET Data

To investigate whether the solvent influenced the specific surface area and pore size of AC, we performed BET tests on the catalyst before and after the reaction (F denotes the fresh catalyst before the reaction, and U denotes the used catalyst after the reaction), as shown in Table 1. The adsorption/desorption isotherms, pore size distributions, and the ratio of mesopores to the total pores of the catalysts are shown in Figure 6. According to the BET data, the specific surface area before the reaction was 903.36 m^2^/g, the pore volume was 0.32 cm^3^/g, and the pore diameter was 2.70 nm when pure water was used as the solvent; when AC was used and the treatment conditions were the same, the specific surface area increased to 991.49 m^2^/g after replacing the solvent with mixed solvent A, and the pore volume was 0.36 cm^3^/g with a pore size of 2.69 nm, indicating that changing the solvent reduced the particle dispersion and avoided the reduction in specific surface area due to pore blockage [28]. This also corroborates with the results of TEM characterization.

Increasing the catalyst-mediated pore ratio is beneficial for the preparation of the Vac reaction using the acetylene method [29,30]. As shown in Figure 6b, the replacement of the solvent improved the catalyst-mediated pore ratio from 60.2% to 61.3%.

### 3.6. TPD Data Analysis

The adsorption process of reactant molecules is closely related to the catalytic performance of the catalyst. Therefore, we characterized the catalyst for CH_3_COOH-TPD, where the desorption area corresponds to the amount of CH_3_COOH adsorbed, and the desorption intensity corresponds to the desorption temperature [31]. In contrast, for the acetylene synthesis of the Vac reaction, the enhancement of acetic acid adsorption is beneficial to the reaction [23,32,33]. On the one hand, it is because of the acetic acid adsorption mechanism [34]. On the other hand, it is because acetylene reacts with CH_3_COOH on Zn(OAc)_2_. As shown in Figure 7, the catalyst prepared using mixed solvent A showed a decrease in desorption temperature by 5 °C and a significant increase in desorption area relative to the catalyst prepared with water as the solvent. Combined with the previous characterization results, we speculate that this is due to the dispersion of the active sites, which leads to an increase in the adsorption of CH_3_COOH and thus an increase in catalyst activity.

## 4. Conclusions

In this study, the solvent in the catalyst was optimized in terms of the nature of the catalyst carrier. The solvent with adjusted polarity bridges the gap between the active component (polar) and the carrier (AC, nonpolar) because the active component is often poorly dispersed on the carrier and can block the pore channels of the catalyst. We found that the decrease in solvent polarity enhanced the loading of the active component of the catalyst; however, solvent polarity that was too low reduced the solubility of the active component and thus inhibited the loading of the active component. Using the optimized solvent, the dispersion of the active component was significantly improved, and the particle size of the active component was significantly reduced, as confirmed by TEM mapping. The solvent also increased the specific surface area of the catalyst and the content of intermediate pores of the active material, which, in turn, affected the adsorption of acetic acid and improved the catalytic performance. This study is somewhat limited in that we only tested the effect of a few common solvents on the catalyst activity and did not perform extensive testing. The prepared catalysts were only applied to the synthesis of vinyl acetate and not to other reactions. Solvent plays an important role in the process of synthesizing industrial catalysts and its optimization is widely relevant for the synthesis of industrial catalysts using the impregnation method.

## Figures and Tables

**Figure 1 nanomaterials-13-00393-f001:**
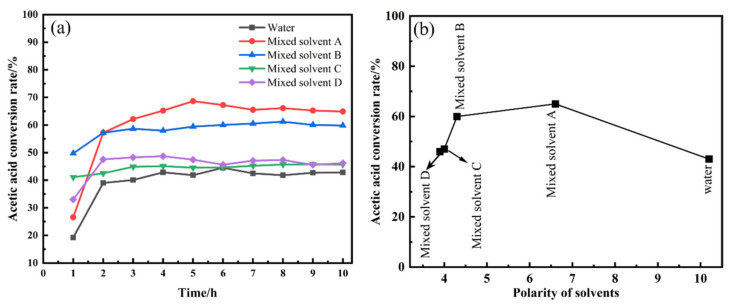
(**a**) Acetic acid conversion using different solvents for catalyst preparation; (**b**) acetic acid conversion as a function of the polarity of the solvent used to prepare the catalyst.

**Figure 2 nanomaterials-13-00393-f002:**
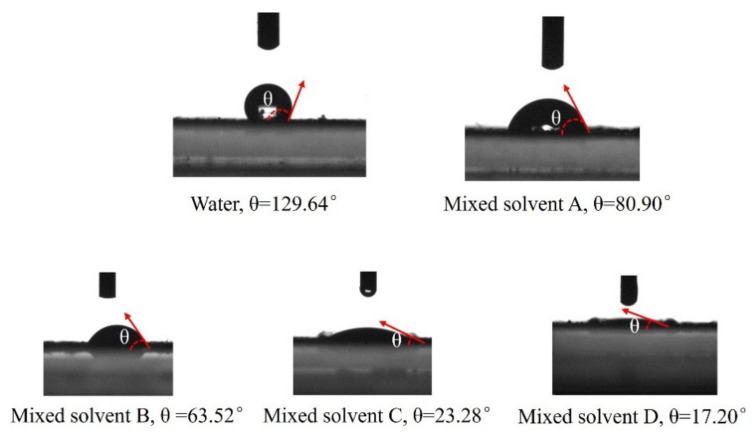
Contact angles of different solvents with AC.

**Figure 3 nanomaterials-13-00393-f003:**
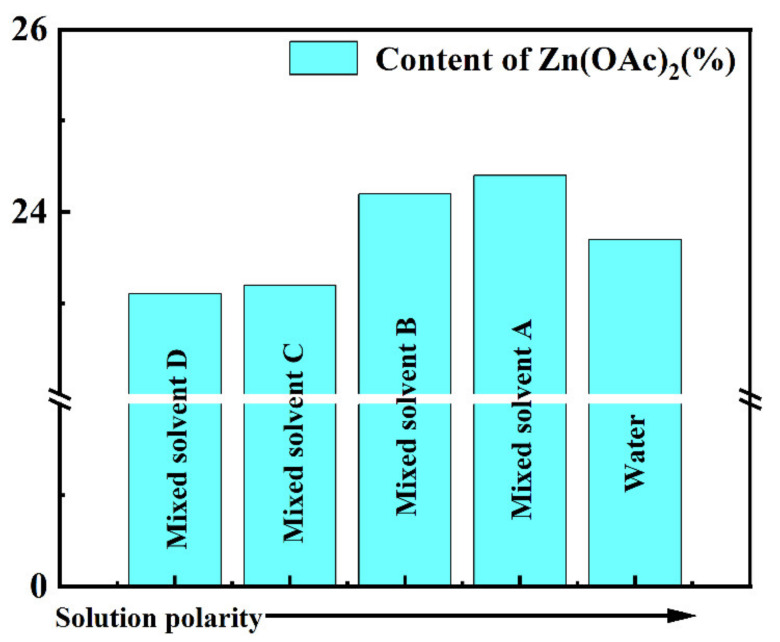
Relationship between polarity and active component loading.

**Figure 4 nanomaterials-13-00393-f004:**
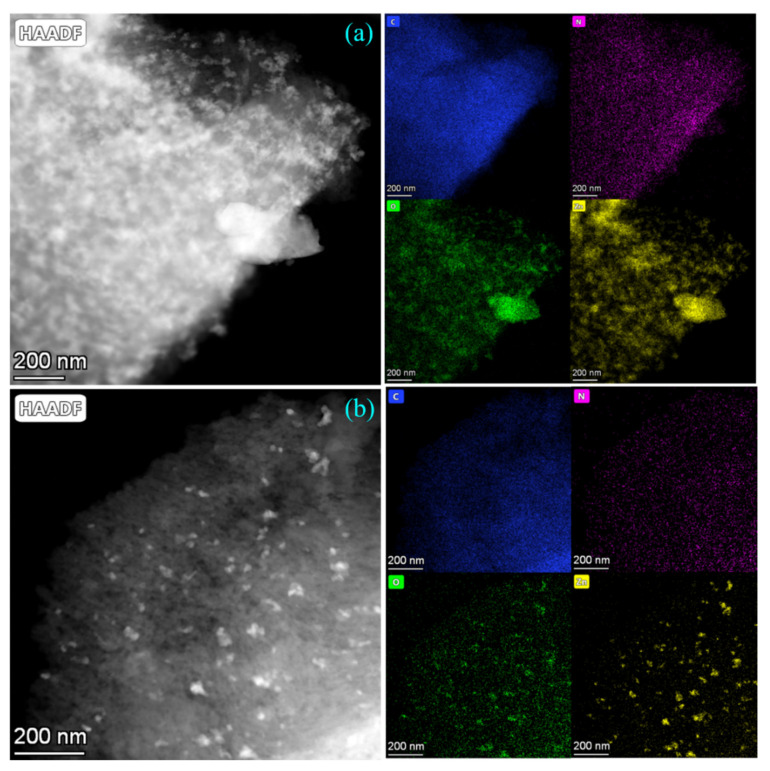
TEM mapping image of the catalyst prepared with water (**a**); TEM mapping image of the catalyst prepared with mixed solvent A (**b**).

**Figure 5 nanomaterials-13-00393-f005:**
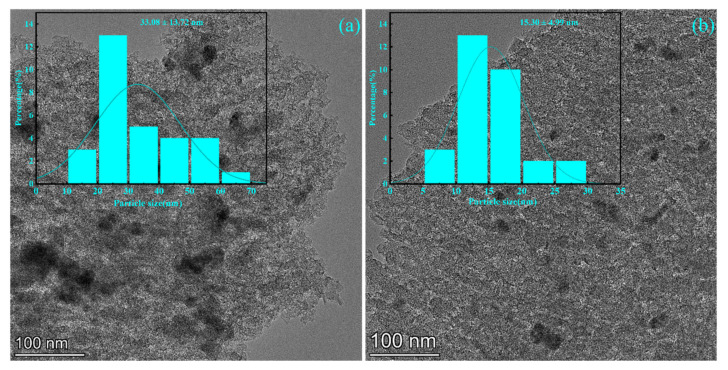
(**a**) TEM image and particle size distribution of the catalyst prepared with water as the solvent; (**b**) TEM image and particle size distribution of the catalyst prepared with mixed solvent A.

**Figure 6 nanomaterials-13-00393-f006:**
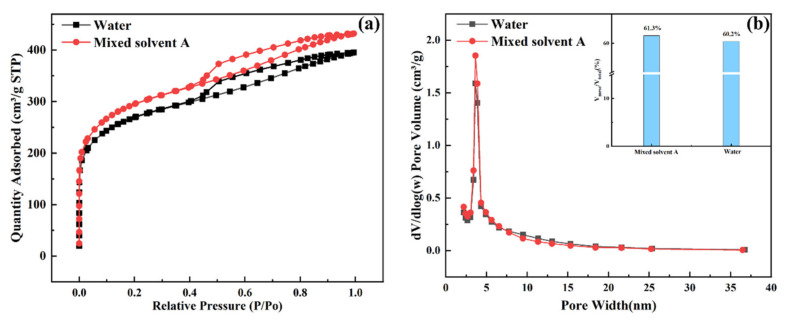
Catalyst adsorption/desorption isotherms (**a**); catalyst pore volume pore size distribution and distribution of mesopores to total pore volume (**b**).

**Figure 7 nanomaterials-13-00393-f007:**
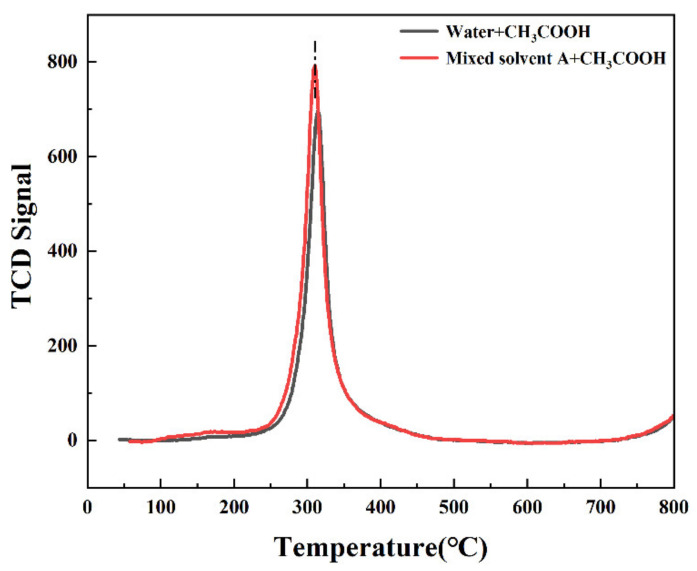
TPD results of the catalysts prepared using water and mixed solvent A as the solvents.

**Table 1 nanomaterials-13-00393-t001:** Specific surface area, pore volume, and pore size of each sample.

Samples	BET Surface Area m^2^/g	Pore Volume cm^3^/g	Pore Size nm
F-water	903.36	0.32	2.70
U-water	388.53	0.25	3.39
F-mixed solvent A	991.49	0.36	2.69
U-mixed solvent A	601.44	0.29	3.02

## Data Availability

Data sharing not applicable. No new data were created or analyzed in this study. Data sharing is not applicable to this article.

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
