# Peer review of "Solvent Effects in the Preparation of Catalysts Using Activated Carbon as a Carrier"

_nanomaterials, 2023, doi:10.3390/nano13030393_

Round 1
Reviewer 1 Report
Specific Comments
1. Abstract: Abstract should be checked and revised carefully with briefly introducing the work plan and key findings.
2. The author should include the Surface area results in the abstract.
3. Keywords contain less. Add more keywords
4. In the introduction need more details. Add more details of Activated carbon and add more recent references.
5. In the introduction part add more literature survey of different activated carbon.
6. The author should provide FTIR and XRD results. It’s useful for the readers to identify the surface functional groups.
7. The authors need to cite some more references in the following section, 3.1, 3.2, 3.3, 3.4, 3.5, and 3.6.
8. Conclusion: Strengthen the section by adding novelty, limitations, and implications of the study.
9. Grammar and typos. The manuscript contains some grammatical and typographical errors. The authors need to thoroughly revise the manuscript and correct the errors.
Reviewer 2 Report
The present article evidences experimentally the role of solvent (using mixtures with different polarities) for the preparation of Zinc acetate supported on activated carbons and their performance for vinyl acetate shynthesis from acetic acid and acetylene as a test reaction. However, in my opinion, the discusion of the results is sometimes confusing and not clear. I do recomed publication, after major revision.
- Line 49: References should be provided in this point.
- Line 58: Carbon powder or activated carbon powder are mentioned in the materials. Which one is used as catalyst support?
- Line 74: If you are using activated carbon as supports, why the mixture of zinc acetate and activated carbon is carbonized at 800ºC?.
- Line 79: Did you used XPS for the characterization of the samples?. Which are the results obtained?
- Line 85: References should be provided.
- Line 105: “The products were tested and analyzed usin gas chromatography”. However, only acetic acid conversion is provided. In addition to vinyl acetate, which other products are obtained?
- Line 133: How were the contact angles of the solvents meassured?
- Line 179: TEM images and particle size distributions should be mentioned in the title of the figure.
- Regarding the textural parameters of the used catalysts, is coke being deposited during the reaction?. Why the reduction of BET Surface area is significantly lower for U-Water sample?.
- Line 212: TPD data analysis: The readers who work with carbon-based materials would expect the evolution of CO, CO2, H2O and H2 as a function of temperature during the TPD for the Zn acetate supported activated carbons. Carbon-oxygen groups of acidic character (carboxylic, lactonic) evolve as CO2 upon thermal decomposition in a typical TPD analysis, whereas non-acidic ones (carbonyl, ether, quinone and phenol) evolve as CO. Anydride surface groups evolve as both CO and CO2. How do you know that acetic acid is the only component being desorbed within that temperatura range if you are using a TCD? In my opinion, if you want to measured only the acetic acid desorption profile another detector should be used. Instead, the evolution of CO and CO2 should be provided while measuring a TPD.
Round 2
Reviewer 2 Report
The revised version has been improved and my suggestions and questions were answered satisfactory.